# Normal embryonic development and neonatal digit regeneration in mice overexpressing a stem cell factor, *Sall4*

**Katherine Q. Chen**[1], **Aaron Anderson**[1], **Hiroko Kawakami**[1,2,3], **Jennifer Kim**[1], **Janaya Barrett**[1], **Yasuhiko Kawakami**[1,2,3] *

**1** Department of Genetics, Cell Biology and Development, University of Minnesota, Minneapolis, Minnesota, United States of America, **2** Stem Cell Institute, University of Minnesota, Minneapolis, Minnesota, United States of America, **3** Developmental Biology Center, University of Minnesota, Minneapolis, Minnesota, United States of America

* kawak005@umn.edu

## Abstract

*Sall4* encodes a transcription factor and is known to participate in the pluripotency network of embryonic stem cells. *Sall4* expression is known to be high in early stage post-implantation mouse embryos. During early post-gastrulation stages, *Sall4* is highly expressed in the tail bud and distal limb buds, where progenitor cells are maintained in an undifferentiated status. The expression of *Sall4* is rapidly downregulated during embryonic development. We previously demonstrated that *Sall4* is required for limb and posterior axial skeleton development by conditional deletion of *Sall4* in the *T* (*Brachyury*) lineage. To gain insight into *Sall4* functions in embryonic development and postnatal digit regeneration, we genetically overexpressed *Sall4* in the mesodermal lineage by the *TCre* transgene and a novel knockin allele of *Rosa26-loxP-stop-loxP-Sall4*. In significant contrast to severe defects by *Sall4* loss of function reported in previous studies, overexpression of *Sall4* resulted in normal morphology and pattern in embryos and neonates. The length of limb long bones showed subtle reduction in *Sall4*-overexpression mice. It is known that the digit tip of neonatal mice has level-specific regenerative ability after experimental amputation. We observed *Sall4* expression in the digit tip by using a sensitive *Sall4-LacZ* knock-in reporter expression. *Sall4* overexpression did not alter the regenerative ability of the terminal phalange that normally regenerates after amputation. Moreover, *Sall4* overexpression did not confer regenerative ability to the second phalange that normally does not regenerate after amputation. These genetic experiments show that overexpression of *Sall4* does not alter the development of the appendicular and axial skeleton, or neonatal digit regeneration. The results suggest that *Sall4* acts as a permissive factor rather than playing an instructive role.

## Introduction

*Sall4* encodes a zinc finger transcription factor [1, 2], and mutations in human *SALL4* cause Duane-radial ray syndrome, an autosomal dominant disorder [3, 4]. Patients with this

---

**Data Availability Statement:** All relevant data are within the manuscript.

**Funding:** This study was supported by a grant from the National Institutes of Health (https://www.

nih.gov) to YK (R01AR064195). The funders had no role in the study design, data collection and analysis, decision to publish, or preparation of the manuscript.

**Competing interests:** The authors have declared that no competing interests exist.

syndrome exhibit various defects, including upper limb deformities, aberrant ocular movements, renal agenesis, unilateral deafness, choanal atresia, external ear malformations, and ventricular septal defects of varying degrees [3, 5]. *Sall4*$^{+/-}$ mice partially recapitulate human patients' symptoms, such as anorectal and heart anomalies [6–8]. These reports demonstrate that *Sall4/SALL4* is necessary for the development and function of a variety of tissues and organs.

*Sall4* expression has been investigated in pluripotent stem cells and mouse embryos. *Sall4* expression is detected in early cleavage stages [8–10]. In blastocysts, *Sall4* is expressed in the inner cell mass and trophectoderm [6, 9], and contributes to the proliferation of cells in the inner cell mass of blastocysts [6]. In pluripotent mouse embryonic stem (ES) cells, SALL4 binds to AT-rich sequences and represses activities of differentiation-promoting enhancers [11, 12]. After implantation, *Sall4* is widely expressed until embryonic day E7.5 [13, 14]. After gastrulation, *Sall4* is highly expressed in the posterior part of the body, such as the tail bud and in the presomitic mesoderm, where undifferentiated progenitor cells undergo proliferation and differentiation at E9.5 –E12.5 [13–15]. *Sall4* is also expressed in the distal mesenchyme of developing limb buds, where cells are maintained undifferentiated by the signals from the apical ectodermal ridge [13, 15, 16]. In the postnatal stages, *Sall4* is expressed in undifferentiated spermatogonia in the testis [17–19], and is essential for the maintenance of undifferentiated spermatogonia [20]. These reports collectively show that *Sall4* expression is associated with undifferentiated cells and tissues.

The functions of *Sall4* in mouse development have also been investigated. *Sall4* null mouse embryos arrest at the peri-implantation stage, indicating its presence is required in the pluripotent epiblast [6, 9]. Conditional deletion of *Sall4* in the meso-endoderm lineage and neuromesodermal progenitors using *TCre* (or *Brachyury-Cre*) caused tail truncation through early depletion of progenitor cells [21]. The *TCre*; *Sall4* mutants also show defects of the proximal-anterior skeleton in the hindlimb and disorganized vertebrate, posterior to the lumber level [15, 21]. These studies suggest that *Sall4* is required for maintenance of undifferentiated progenitor cells and their differentiation in mouse embryos.

In postnatal stages, *Sall4* is involved in regeneration in non-mammalian animals. In the axolotl, *Sall4* plays a critical role in scar-free cutaneous wound healing through the regulation of collagen gene transcription [22]. In the regenerating limbs of *Xenopus* tadpoles, *Sall4* is detected in the blastema [23, 24]. It has been suggested that *Sall4* maintains cells in an undifferentiated state in the blastema and regulates patterning of the regenerating limb at later stages [23, 25]. In the case of mice, it has been demonstrated that fetus, neonates, and adults possess the ability to regenerate digit tips [26, 27]. The regenerative ability is level specific and restricted to the terminal phalange [28, 29]. However, it remains unknown whether *Sall4* is expressed in the mouse digit tip and if *Sall4* plays a role in digit tip regeneration in mice.

In addition to expression and function in mouse development and regeneration, *SALL4* expression is reported in many types of human cancers (reviewed in [30, 31]). *SALL4* expression is normally absent in most adult tissues, but is activated in various human cancers. These cancers include leukemia, germ cell tumors, hepatocellular carcinoma, gastric cancer, colorectal carcinoma, esophageal squamous cell carcinoma, breast cancer, endometrial cancer, lung cancer, and glioma [30, 31]. Studies suggest that *SALL4* functions in cancer survival, drug resistance, and metastasis. Furthermore, detailed literature review suggests that *SALL4* expression increases both mortality and recurrence of cancer, and suggests *SALL4* as an important prognostic marker [32].

*Sall4* expression patterns and functional studies suggest that *Sall4* is necessary in various tissues and organs during embryonic development, in adult germ cells, and contributes to pathological conditions such as cancer. However, the functions of *Sall4* in tissue/organ development

remain largely unknown in post-implantation mouse embryos due to the early lethality of *Sall4* null mutants [6–9]. To gain insight into *Sall4* functions, we genetically overexpressed *Sall4* by recombination using the *TCre* transgene. In significant contrast to severe defects in reported *Sall4* loss of function studies, our results show that *Sall4* overexpression in the *T*-lineage results in the normal development of mouse embryos and no alteration of digit tip regenerative ability.

## Materials and methods

### Generation of the inducible *Sall4* knock-in allele and mice breeding

The inducible *Sall4* expression vector was generated as previously described [33] with replacement of *Dmrt1* by the *Flag-Sall4a-T2A-HA-Sall4b* sequence. The expression cassette was knocked into the *Rosa26* locus by homologous recombination in R1 mouse ES cells, as diagrammed in Fig 1. The correctly targeted ES cells were injected into C57BL/6 blastocysts after confirming the normal karyotype, and chimeras were generated. Germline transmission was confirmed by breeding chimeras with C57BL/6 mice and genomic PCR of offspring, and the mouse line was termed as *R26-Sall4*. The *R26-Sall4* mice were bred with C57BL/6 mice at least 3 generations before experimentation.

C57BL/6 and CD-1 mice were obtained from the Jackson Laboratory and Charles River, respectively. *TCre* mice were previously described [34] and backcrossed >12 generations with C57BL/6. For embryo experiments, *R26-Sall4* mice were timed mated with *TCre* mice. For postnatal skeletal analysis and digit regeneration experiments, *R26-Sall4* mice were bred with *TCre* mice. Experimental protocols were approved by the University of Minnesota Animal Care and Use Committee (protocol number: 1706-34894A, 1910-37482A).

### ES cell screening and mouse genotyping

To screen targeted ES cells, a DNA fragment described in [33] was labelled using DIG DNA labeling mix (Roche, Cat#11277065910) and genomic Southern hybridization was performed following a standard procedure [35]. Genomic PCR of ES cell DNA and PCR genotyping on tail clip DNA for the *R26-Sall4* mice were conducted as previously described [33].

### Skeletal preparation and whole mount in situ hybridization

For alcian blue/alizarin red staining, neonates or the postnatal day (P) 21 autopod were fixed in 50% ethanol. Skeletal staining and whole mount in situ hybridization of embryos were performed as previously described [15, 21].

### Bone length measurement and statistical analysis

The length of the humerus and femur was measured by using FIJI. Lines were drawn parallel to the long axis of each bone, then perpendicular to those lines such that the lines touched the edges of the bone. The length of the bone was measured by using the distance between the edges of the line parallel to the long axis. The measurement was performed by two independent individuals, and the average value was used for statistical examination, which was performed by One-way ANOVA using JMP software version 16.

### Neonatal digit amputation

Neonatal (P0) mice were anesthetized on ice and the digits of the hindlimbs were amputated [36]. The regenerative amputations were performed using microdissection scissors (Fine

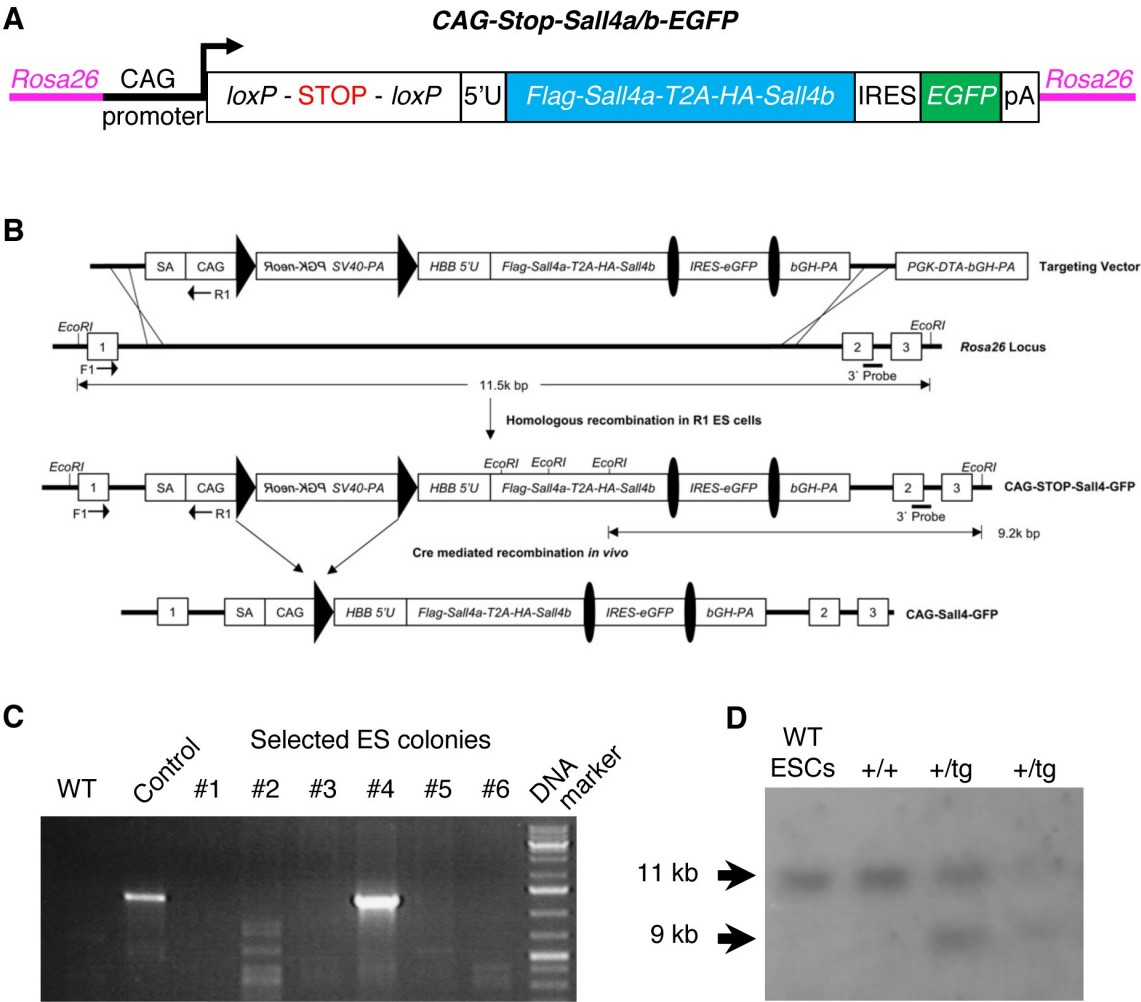

**Fig 1. *Sall4* conditional expression transgene.** (A) Targeting vector used to insert the conditional expression construct for *Sall4* into the *Rosa26* locus in ES cells. (B) Schematic of the targeting strategy. The targeting vector also contains a diphtheria toxin negative selection cassette (*PGK-DTA-bGH-PA*) with diphtheria toxin expressed from the PGK promoter and followed by a bovine growth hormone poly-adenylation site. Homologous recombination into the *Rosa26* locus removed *PGK-DTA-bGH-PA* and produced the single copy transgene CAG-Stop-*Sall4-IRES-EGFP*. Cre-mediated recombination deletes the "floxed" stop cassette to generate *CAG-Sall4-IRES-EGFP*. (C) Correct homologous recombination of the 5' homology arm of the targeting vector into *Rosa26* was confirmed by PCR of ES cell genomic DNA using primers F1 and R1 (indicated in panel B), which amplify between exon 1 of *Rosa26* and the CAG promoter. No amplification product was detected in wild-type ES cells, but the predicted 1248 bp amplicon was detected in a targeted ES cell clone (#4). Control DNA was prepared from the tail clip of conditional *Dmrt1* expression mouse that was generated using the same knockin system (33). (D) Homologous recombination of the 3' vector arm was confirmed by Southern blotting using the 3' probe indicated in panel B. Wild-type ES cell DNA generated in detection of the predicted 11 kb fragment, whereas targeted clones (two right lanes) generated both the 11 kb fragment and a 9 kb fragment corresponding to the targeted allele.

Science Tools, Cat#15000–00) cross the distal third of the terminal phalange [27]. For non-regenerative amputation, the distal third of the second phalange was amputated.

## Western blotting

For embryos, E10.5 embryos were dissected, and EGFP-positive (*TCre^{+/tg}; R26-Sall4^{+/tg}*) or negative (*TCre^{+/+}; R26-Sall4^{+/tg}*) embryos were grouped. Embryos were cut at the anterior to the forelimb buds, and the body was used for protein extraction using RIPA buffer with a

protease inhibitor cocktail (Thermo, Cat# 1862209). The extract was centrifuged at 14,000g for 10 min at 4˚C, and the supernatant was used for Western blotting.

For neonatal digits, the entire digits were combined, protein was extracted similar to the E10.5 embryos, and the lysate was subjected to Western blotting.

Primary antibodies against SALL4 (Abcam, ab29122, 1:1000) and GAPDH (Thermo, AM4300, 1:2000) were used. HRP-labelled goat anti-rabbit IgG (BioRad, 170–6515, 1:3000) and anti-mouse IgG (BioRad, 170–6516, 1:3000) were used as secondary antibodies and the signals were detected by a Chemiluminescence system (Thermo, #34080). Signal intensities of bands were calculated by FIJI.

## Results

### Generation of a recombination-dependent *Sall4* overexpression mouse line

In order to test whether continued expression of *Sall4* interferes with normal embryonic development, we sought to express *Sall4* by Cre-dependent genetic recombination. We knocked in a *Sall4* cassette into the *Rosa26* locus as previously described [33]. The *Sall4* transgene expression is silenced by the stop sequence in front of the *Sall4* transgene, which is flanked by the loxP sequences. By Cre-dependent deletion of the stop sequence, *Sall4* is expressed under a strong CAG promoter (Fig 1A and 1B). There are two *Sall4* isoforms, *Sall4a* and *Sall4b*, which are generated through an internal splicing in exon 2 [37]. A previous study in mouse ES cells provided evidence that SALL4A and SALL4B have overlapping but non-identical binding sites [37]. Because functional differences between SALL4A and SALL4B during mouse development are unknown, we expressed both *Sall4a* and *Sall4b*. We inserted the *Sall4a*-T2A peptide sequence-*Sall4b* in the expression cassette, followed by the internal ribosome entry sequence and the EGFP sequence. Cre-dependent recombination causes expression of both *Sall4a* and *Sall4b*, as well as an EGFP reporter. This expression cassette is knocked into the *Rosa26* locus in the R1 mouse ES cells [35]. After electroporation, the ES cells were selected by G418 treatment. Among the 64 clones isolated after G418 selection, six properly targeted embryonic stem cell colonies were confirmed by genomic Southern blotting and genomic PCR (Fig 1C and 1D). After confirming normal karyotype, chimeric mice were generated by injecting one ES cell clone into blastocysts, which were bred with C57BL/6 mice. Germline transmission was confirmed by genomic PCR of offspring. This study was performed using this one strain.

### *Sall4* overexpression in the *Brachyury* lineage results in development of normal skeletal pattern

*Sall4* expression is rapidly downregulated in post-gastrulating stages, and the high level of expression is confined in the distal limb buds and the tail buds [13, 14]. *Sall4* deletion in the *Brachyury* lineage using *TCre* caused severe defects in the appendicular and posterior axial skeleton [15, 21]. In order to test whether continued expression of *Sall4* affects those skeletons, we overexpressed *Sall4* in the *Brachyury* lineage by crossing *R26-Sall4* mice with *TCre* mice.

At E9.5, strong *Sall4* expression is detected in the caudal part of the body and the distal part of the forelimb bud in control embryos (Fig 2A–2C, n = 3). In *TCre; R26-Sall4* embryos, strong *Sall4* expression was broadly detected in the trunk mesoderm, the entire forelimb buds and in the caudal part of the body (Fig 2D–2F, n = 3). The tissue with high levels of *Sall4* expression correlates with the tissue with the EGFP signal that reports recombination and overexpression of the transgenes (Fig 2G and 2H, n = 3). Western blotting with an extract from the embryo (posterior to the anterior-edge of the forelimb bud) also showed an increased level of SALL4 in *TCre*$^{+/tg}$*; R26-Sall4*$^{+/tg}$, compared with *TCre*$^{+/+}$*; R26-Sall4*$^{+/tg}$ embryos. Specifically, the data

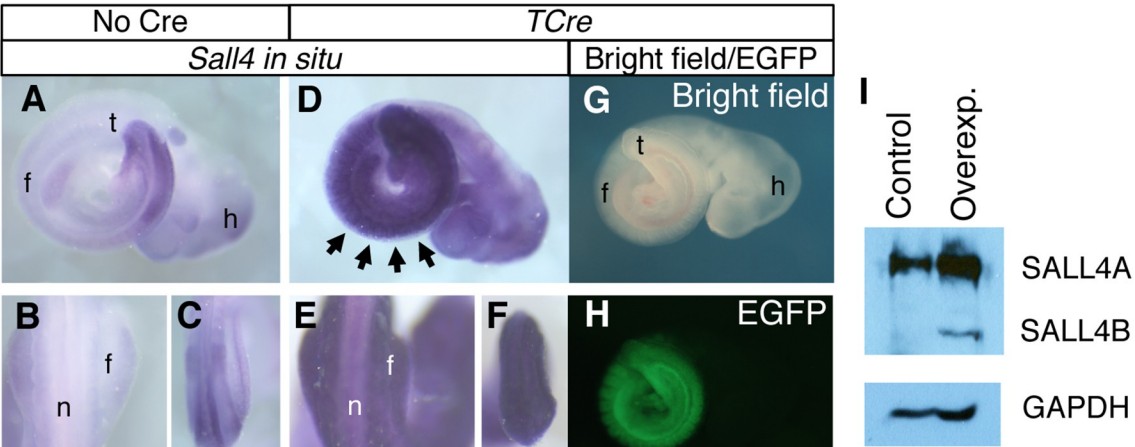

**Fig 2. *TCre*-dependent overexpression of *Sall4*.** (A-C) Endogenous expression pattern of *Sall4* at E9.5. A lateral view of a whole embryo (A), a dorsal view at the forelimb level (B) and a dorsal view of the caudal part of the embryo (C) are shown. (D-F) *Sall4* expression in the *TCre⁺/ᵗᵍ; R26-Sall4⁺/ᵗᵍ* embryo. A lateral view shows strong expression of *Sall4* in the trunk (D). A dorsal view at the forelimb level shows strong expression in the somite and forelimb bud (E). A dorsal view of the caudal part of the embryo shows strong expression in this region (F). (G, H) Lateral views of the *TCre⁺/ᵗᵍ; R26-Sall4⁺/ᵗᵍ* embryo. A bright field image (G) and an EGFP image (H) are shown. The EGFP signal is detected in the trunk, similar to strong *Sall4* expression in D. (I) Western blotting of lysate from *TCre⁺/⁺; R26-Sall4⁺/ᵗᵍ* embryos (control) and *TCre⁺/ᵗᵍ; R26-Sall4⁺/ᵗᵍ* embryos (overexp.). Stronger bands for SALL4A and SALL4B are detected with extracts from the *TCre⁺/ᵗᵍ; R26-Sall4⁺/ᵗᵍ* embryo. Abbreviations. f: forelimb bud, h: head, n: neural tube, t: tail bud.

show that SALL4A overexpression is greater than SALL4B overexpression, while SALL4B expression is increased from baseline in the control. The ratio of SALL4A/GAPDH signal intensity was increased 1.7-fold in *TCre⁺/ᵀᵍ; R26-Sall4⁺/ᵗᵍ* embryos, compared with *TCre⁺/⁺; R26-Sall4⁺/ᵗᵍ* embryos. Taken together, these results show that *Sall4* is overexpressed by *TCre*-dependent recombination.

To assess whether continued *Sall4* overexpression affects skeletal patterns, we stained the skeleton of neonates with alcian blue (cartilage) and alizarin red (bone). Our previous study showed that, in *TCre; Sall4* cKO mutants, the lumber vertebrae were disorganized and the tail was truncated [21]. The hindlimb skeletal elements were severely small or absent, while the forelimb skeleton looked normal [15]. Compared to the defects in *TCre; Sall4* cKO, we did not detect differences in skeletal development between the control (*TCre⁺/⁺; R26-Sall4⁺/⁺*) and the overexpression (*TCre⁺/ᵗᵍ; R26-Sall4⁺/ᵗᵍ*) neonates (Fig 3A–3H). In the *Sall4*-overexpression neonates, the lumber vertebrae exhibited normal morphology (Fig 3B and 3F), and both fore-limbs and hindlimbs developed normally, compared to control neonates (Fig 3C, 3D, 3G and 3H). In order to evaluate whether *Sall4* overexpression affected skeletal development in more detail, we measured the length of the humerus and femur in the allelic series: *TCre⁺/⁺; R26-Sall4⁺/⁺, TCre⁺/ᵗᵍ; R26-Sall4⁺/⁺, TCre⁺/⁺; R26-Sall4⁺/ᵗᵍ*, and *TCre⁺/ᵗᵍ; R26-Sall4⁺/ᵗᵍ*. Statistical examination by One-way ANOVA indicated that there was slight reduction of long bone length in *Sall4*-overexpression neonates (Fig 3I and 3J). The humerus and femur exhibited 8.8% and 4.7% shorter, respectively, of wild-type littermates (*TCre⁺/⁺; R26-Sall4⁺/⁺*). These results indicated that overexpression of *Sall4* does not affect bone pattern and has a negligible effect on the long bone length. Given that the bone length between *TCre⁺/⁺; R26-Sall4⁺/ᵀᵍ* and *TCre⁺/ᵀᵍ; R26-Sall4⁺/ᵀᵍ* did not show significant differences, the presence of the *Sall4* cassette may mildly affect limb development.

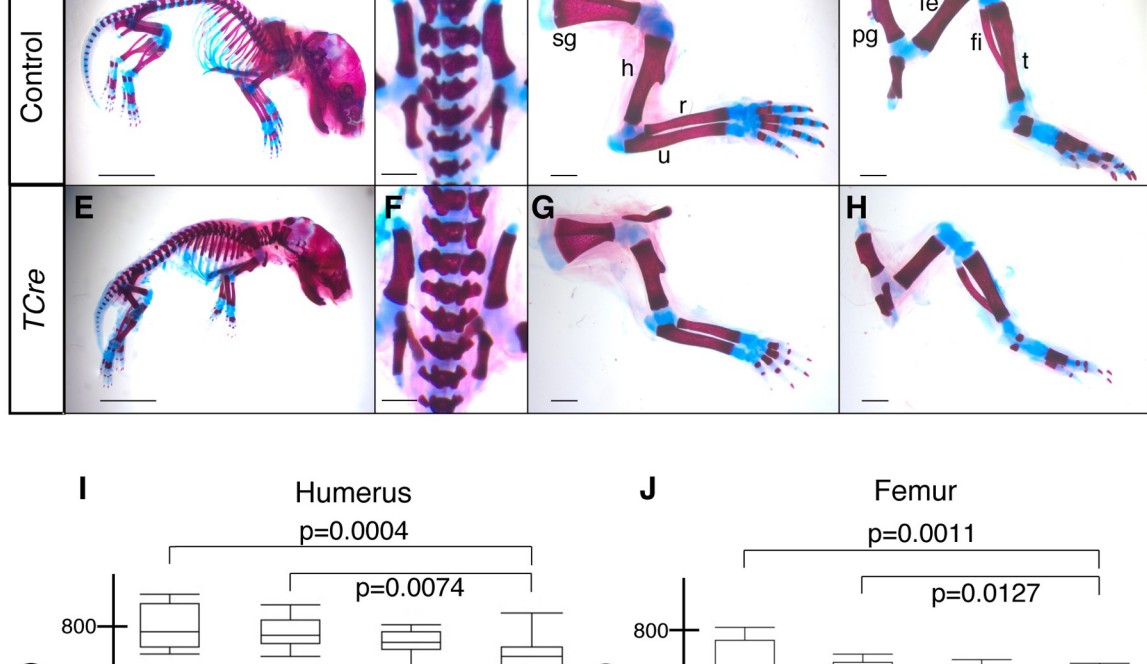

**Fig 3. Analysis of neonatal skeletal development after overexpression of *Sall4* in the *T* lineage.** Skeleton of wild-type (A-D, *TCre*[+/+]; *R26-Sall4*[+/+]) and *Sall4*-overexpression (E-H, *TCre*[+/tg]; *R26-Sall4*[+/tg]) neonates. Lateral views of the entire body (A, E), dorsal views of the lumber region (B, F), forelimbs (C, G) and hindlimbs (D, H) are shown. Scale bar, 5 mm in A and E, 1 mm in B-D, F-H. (I, J) Box plot of the length of the humerus (I) and femur (J) of neonates with the indicated genotypes. WT and Tg indicate wild type and the +/Tg genotype, respectively. Sample numbers are indicated in the panel. Abbreviations. fe: femur, fi: fibula, h: humerus, pg: pelvic girdle, r: radius, sg: shoulder girdle, t: tibia, u: ulna.

## *Sall4* is expressed in the digit tip

Next, we sought to determine whether *Sall4* is expressed in the digit tip, which regenerates after experimental amputation [27]. Because endogenous *Sall4* expression is rapidly downregulated in developing embryos and becomes difficult to detect, we made use of a sensitive LacZ reporter, knocked into the *Sall4* locus [6]. By whole mount LacZ staining, we observed strong *Sall4-LacZ* reporter expression in the tail tip and genital organs at E14.5 (blue arrowhead and red arrowhead, respectively, Fig 4A, n = 4), as previously reported [14]. We also observed signals at the digit tip (black arrows, Fig 4A). The staining at the digit tip was observed at E16.5 (Fig 4B, n = 3, Fig 4D, n = 3) and in neonates (Fig 4C, n = 3, Fig 4E, n = 3). These data show *Sall4* expression in the digit tip during late gestation to neonatal stages.

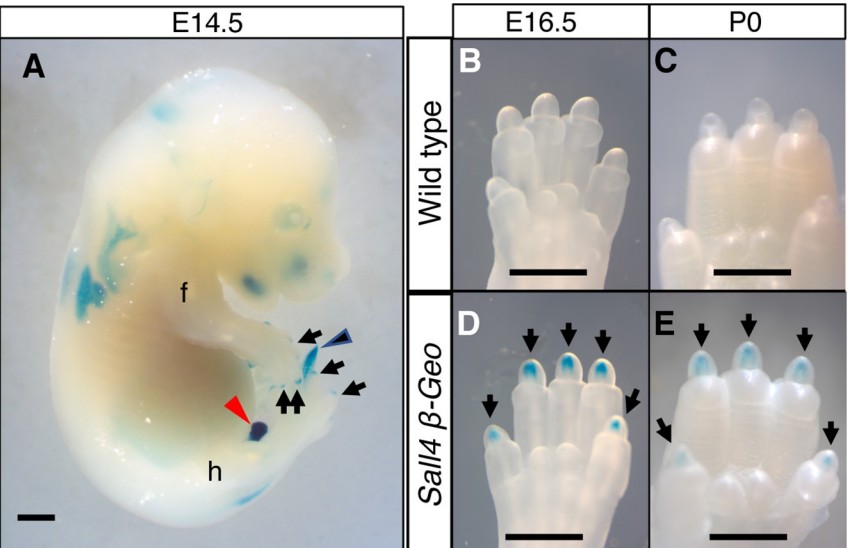

**Fig 4. *Sall4* expression at the digit tip.** (A) Whole-mount LacZ stained *Sall4+/LacZ* embryo at E14.5. Blue and red arrowheads point to signals at the tail tip and genital organ, respectively. Arrows point to signals at the digit tip. Scale bar, 1 mm. (B, C) Ventral views of the wild-type digit tip of the hindlimbs after lacZ staining at indicated stages. (D, E) Ventral views of the *Sall4+/LacZ* digit tip of the hindlimbs after lacZ staining at indicated stages. Arrows point to the LacZ signals. Scale bar, 1 mm in A, 0.2 mm in B-E.

## *Sall4* overexpression does not alter the regenerative ability of the digit tip

The neonatal mouse digit regenerates after amputation at the terminal phalange level. In contrast, the digit fails to regenerate after amputation at the second phalange level [27]. We sought to investigate whether overexpression of *Sall4* can alter regenerative ability of the digit tip. We first confirmed overexpression of SALL4 in the digit of *TCre+/tg; R26-Sall4+/tg* neonates with Western blotting. Tissue lysate prepared from the entire digits of *TCre+/+; R26-Sall4+/tg* neonates did not show detectable SALL4. In contrast, digit lysate from *TCre+/tg; R26-Sall4+/tg* neonates showed overexpressed SALL4A and SALL4B (Fig 5A). We amputated the digit tip at the terminal phalange of control (*TCre+/Tg; R26-Sall4+/+*, n = 7) and *Sall4*-overexpression (*TCre+/*

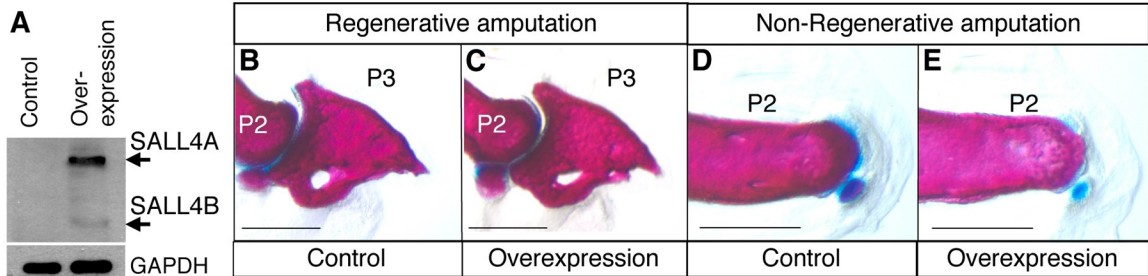

**Fig 5. *Sall4* overexpression does not alter the regenerative ability of the digit tip.** (A) Western blotting of lysate from *TCre+/+; R26-Sall4+/Tg* and *TCre+/tg; R26-Sall4+/tg* neonatal digits. Arrows point to SALL4A and SALL4B bands. (B, C) Alcian blue/Alizarin red-stained digits from 21 day-old control (*TCre+/tg; R26-Sall4+/+*) and *Sall4*-overexpression (*TCre+/tg; R26-Sall4+/tg*) mice, amputated at the tip of the terminal phalange level at the P0 stage. (D, E) Alcian blue/Alizarin red-stained digits from 21 day-old control (*TCre+/tg; R26-Sall4+/+*) and *Sall4*-overexpression (*TCre+/tg; R26-Sall4+/tg*) mice, amputated at the second phalange level at the neonatal stage. Scale bar: 500 μm in B-E. P2 and P3 in panels B-E indicate the second phalange and the terminal phalange, respectively.

*tg*; *R26-Sall4*<sup>+/tg</sup>, n = 9) neonates, and stained the digits with alcian blue/alizarin red at day 21. The regenerated digit tip skeleton was comparable between the two groups (Fig 5B and 5C). This result indicates that overexpression of *Sall4* did not impair regenerative ability of the digit tip. Next, we amputated at the digit at the second phalange level at the neonate stage. In both control (n = 7) and *Sall4*-overexpression (n = 9) mice, alcian blue/alizarin red staining showed no regeneration of the amputated digits (Fig 5D and 5E). This result indicates that overexpression of *Sall4* does not supply regenerative ability to non-regenerative digits. Overall, these results indicate that *Sall4* overexpression does not change regenerative and non-regenerative ability of the neonatal mouse digit tip.

## Discussion

Several studies, including ours, investigated functions of *Sall4* in animal development. *Sall4*<sup>-/-</sup> mouse embryos die at the peri-implantation stage, indicating that *Sall4* is required for epiblast survival [6, 9]. Due to early lethality of *Sall4*<sup>-/-</sup> embryos, studies with mice were performed by conditional gene inactivation, which identified that *Sall4* was required in the development of the neuromesodermal progenitors [21], limb progenitors [15, 38], primordial gem cells [39], oocytes [10], and spermatogonia [17]. In addition, the *Sall4*<sup>+/-</sup> allele or *Sall4* gene-trap alleles were used to investigate genetic interaction between *Sall4* and other genes. For example, *Sall1*<sup>+/-</sup>; *Sall4*<sup>+/-</sup> mouse embryos do not develop the kidney [6]. The *Sall4*<sup>+/gene-trap</sup> background enhances neural tube defects in *Sall2*<sup>-/-</sup> mice [40]. Furthermore, analysis of *Tbx5*<sup>+/-</sup>; *Sall4*<sup>+/gene-trap</sup> embryos demonstrated their interaction for forelimb and heart patterning [16]. In zebrafish and *Xenopus* systems, studies using morpholino-based knockdowns, and more recently, a CRISPR-Cas9-based genetic knockout approach in zebrafish were conducted [41]. These studies identified a *sall4* requirement in pectoral fin development [42], hematopoiesis [43], and posterior neuroectoderm development [44, 45]. In chick embryos, a truncated form of *Sall4*, which might interfere with endogenous *Sall4*, was expressed by electroporation to the otic placode for loss-of-function study [46].

*Sall4* overexpression approach was also conducted to gain insight into *Sall4* function in animal development. For example, full length *Sall4* overexpression by plasmid electroporation into the head ectoderm in chick embryos showed that *Sall4* can induce invagination of the non-placodal head ectoderm [46]. In this system, a region of interest is targeted by plasmid injection and electroporation, and the expression is lost as cells divide and dilute the electroporated plasmid. In another approach, synthetic *sall4* mRNA was injected into blastomeres at the early cleavage stages of *Xenopus* [45]. With this approach, one can overexpress *sall4* in the entire embryos or target specific blastomeres to spatially restrict overexpression. Nonetheless, as cells divide and the injected mRNA is degraded, the overexpressed transcripts are lost over the course of development. In our study, we used Cre-loxP-dependent conditional overexpression of *Sall4* in mouse embryos. A *Sall4* expression cassette is knocked into the *Rosa26* locus with the loxP-stop-loxP sequence, which is commonly performed for conditional overexpression in mice [47, 48]. This genetic recombination approach allows cells to inherit the recombined allele and continue to overexpress *Sall4*.

We chose to use *TCre*, which recombines broadly in meso-endoderm and neuromesodermal progenitors [21, 34]. *Sall4* is broadly expressed in gastrulating embryos, and as discussed above, conditional deletion of *Sall4* in various types of progenitor cells demonstrated that *Sall4* is required in embryo development. Moreover, *Sall4* expression is rapidly downregulated from the limb bud mesenchyme, somites, and other areas of the embryo [14]. However, *Sall4* overexpression did not cause significant developmental defects, and the overall morphology of these embryos and neonates is comparable to wild-type embryos or neonates. The lack of

defects was surprising to us, given the conditional loss of *Sall4* causes various defects during embryonic development. However, a similar observation was also reported in *Xenopus*, in which *Sall4* was overexpressed by injecting mRNA into animal-dorsal cells of 4-cell stage embryos [45]. Compared to experiments by the same group, where *Sall4* loss-of-function by morpholino injection resulted in a change in expression patterns of several neural markers, *Sall4* overexpression did not affect the expression pattern of marker genes in early stage *Xenopus* embryos. In our study, while the overall morphology was not changed by overexpression of *Sall4* in the *T*-lineage, we observed a slight reduction of limb long bone length. The reduction was very subtle, although statistically significant. Based on the comparison of bone length, we speculate that harboring the *Sall4* expression cassette, although its expression is silenced, might have a subtle effect on skeletal growth.

Studies in non-mammalian animals suggested that *Sall4* is involved in limb regeneration. *Sall4* is upregulated during limb regeneration in *Xenopus* [23, 24] and axolotls [25]. It has been hypothesized that *Sall4* keeps cells in an undifferentiated state in the blastema during the early stages of blastema formation [23–25]. Our study showed that *Sall4* is expressed in the digit tips of mouse fetuses and neonates. Mouse digit tips can regenerate after amputation at the terminal phalange level where the nail bed is associated [27]. The expression pattern is consistent with the postulated role of *Sall4* in regeneration. However, our study showed that *Sall4* overexpression does not affect digit tip regeneration. Overexpression of *Sall4* did not interfere with the normal regenerative ability of the digit tip, when amputated at the terminal phalange. Moreover, *Sall4* overexpression did not provide a regenerative ability, when amputated at the distal part of the second phalange level. One possible interpretation of our results is that *Sall4* alone cannot provide limb regenerative ability. *Sall4* may require highly regenerative cellular or tissue context to contribute to limb regeneration. Loss of function experiments in *Xenopus* or axolotls would provide insights into *Sall4* function in limb regeneration.

A lack of evident phenotype after *Sall4* overexpression suggests that SALL4 requires molecular partners in the experimental setting examined in this study. For example, SALL4 interacts with PLZF and DMRT1 in undifferentiated spermatogonial progenitor cells by coimmunoprecipitation experiments [17, 49]. Our recent study also suggests that SALL4 functions, at least in part, depend on molecular partners. More specifically, SALL4 ChIP-seq in mouse ES cells [12] and in mesoderm tissues from the posterior part of E9.5 embryos exhibited significant difference in SALL4-enriched sequences [21]. Moreover, transcription factor motifs in the SALL4-enriched sequences are also distinct between ES cells and the embryonic tissue. These reports suggest that, in addition to directly binding to DNA [11], SALL4 also acts in a protein complex. In the latter case, *Sall4* function may require cell or tissue-specific interacting partners.

Overall, our study in developing embryos and the neonatal digit showed that overexpression of *Sall4* does not alter the developmental and regenerative processes that we examined. These results suggest that *Sall4* acts as a permissive factor rather than an instructive factor in these developmental and regeneration contexts.

Although we did not observe significant effects by overexpressing *Sall4* in the *Brachyury* lineage, our *Sall4* overexpression model may be used in different contexts to gain insights into *Sall4* functions in the future. It is suggested that *Sall4* blocks differentiation of mouse ES cells [11, 12]. It will be interesting to interrogate whether *Sall4* overexpression can alter developmental processes that are not examined in this study, such as neural development. It will also be of interest to test whether *Sall4* overexpression can alter regenerative processes that are not examined in this study, as well as cancer cell phenotypes.

## Supporting information

**S1 Raw images. Western raw images.**
(PDF)

## Acknowledgments

We are grateful to Sandy Zhang for her excellent technical assistance. We are also grateful to Dr. David Zarkower for sharing materials to generate the *R26-Sall4* mouse line, to the University of Minnesota Cytogenomics Laboratory and the Mouse Genetics Laboratory for their excellent service in karyotyping and chimera production, respectively. KQC, AA and JK were partially supported by the University of Minnesota's Undergraduate Research Opportunity Program.

## Author Contributions

**Conceptualization:** Yasuhiko Kawakami.

**Formal analysis:** Katherine Q. Chen, Aaron Anderson, Janaya Barrett, Yasuhiko Kawakami.

**Funding acquisition:** Yasuhiko Kawakami.

**Investigation:** Katherine Q. Chen, Aaron Anderson, Hiroko Kawakami, Jennifer Kim, Janaya Barrett, Yasuhiko Kawakami.

**Project administration:** Yasuhiko Kawakami.

**Supervision:** Yasuhiko Kawakami.

**Visualization:** Yasuhiko Kawakami.

**Writing – original draft:** Katherine Q. Chen, Yasuhiko Kawakami.

**Writing – review & editing:** Katherine Q. Chen, Aaron Anderson, Hiroko Kawakami, Jennifer Kim, Janaya Barrett, Yasuhiko Kawakami.

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
