## [Decision Letter · Decision Letter 0]

18 Feb 2022

PONE-D-22-00272Normal embryonic development and neonatal digit regeneration in mice overexpressing a stem cell factor, Sall4PLOS ONE

Dear Dr. Kawakami,

Thank you for submitting your manuscript to PLOS ONE. After careful consideration, we feel that it has merit but does not fully meet PLOS ONE’s publication criteria as it currently stands. Therefore, we invite you to submit a revised version of the manuscript that addresses the points raised during the review process.

We look forward to receiving your revised manuscript.

Kind regards,

Atsushi Asakura, Ph.D

Academic Editor

PLOS ONE

Journal Requirements:

"KQC, AA and JK were partially supported by the University of Minnesota’s Undergraduate Research Opportunity Program. This study was supported by a grant from the National Institutes of Health to YK (R01AR064195). The funders had no role in the study design, data collection and analysis, decision to publish, or preparation of the manuscript."

We note that you have provided funding information. However, funding information should not appear in the Funding section or other areas of your manuscript. We will only publish funding information present in the Funding Statement section of the online submission form. 

"This study was supported by a grant from the National Institutes of Health (https://www.nih.gov) to YK (R01AR064195). The funders had no role in the study design, data collection and analysis, decision to publish, or preparation of the manuscript."

Reviewers' comments:

Reviewer's Responses to Questions

**Comments to the Author**

1. Is the manuscript technically sound, and do the data support the conclusions?

Reviewer #1: Yes

Reviewer #2: Yes

2. Has the statistical analysis been performed appropriately and rigorously? 

Reviewer #1: Yes

Reviewer #2: Yes

3. Have the authors made all data underlying the findings in their manuscript fully available?

Reviewer #1: Yes

Reviewer #2: Yes

4. Is the manuscript presented in an intelligible fashion and written in standard English?

Reviewer #1: Yes

Reviewer #2: Yes

5. Review Comments to the Author

Reviewer #1: Chen et al. inserted Sall4 into the Rosa26 locus and overexpressed Sall4 in mice via mesodermal Cre-dependent recombination. Although Sall4a and Sall4b were overexpressed in the caudal part of the embryo and the entire forelimb bud, minimal abnormalities were detected in early embryogenesis and digit regeneration. Although this manuscript describes negative results, the included information and the generated mouse strain are useful for the research community.

Major comments

1. The authors should describe the number of homologous recombinant ES clones and the number of independent mouse strains that were generated and analyzed. If this study involved only one ES clone and one mouse strain, those should be mentioned as limitations of the study; such information is necessary for objective interpretation by readers.

2. Figure 2: A description should be provided concerning the number of biologically independent experiments performed with reproducible results. This point applies to all figures in the manuscript.

3. Figure 2I: Because intensities differ among the loading controls (GAPDH bands), it is difficult to judge the quantitative increase in Sall4 protein expression. The ratio of Sall4 band intensity to GAPDH band intensity should be quantified, and the percentage increase should be provided. In addition, the results should be described in greater detail. For example, Sall4a overexpression is greater than Sall4b overexpression, while Sall4b expression is increased from baseline in the control.

4. Figure 3I, J: the definitions of the dots and double dots should be clarified; they also appear to be mislocated in the graphs. Because there are no significant differences between Cre-WT/Rosa-Tg and Cre-Tg/Rosa-Tg, the effect of Sall4 overexpression is negligible. Instead, the presence of Sall4 cassette itself may mildly affect limb development. Such interpretation should be provided in the Results section, not the Discussion section.

5. The Discussion section largely consists of repetitive descriptions of the results; it also contains many descriptions of loss-of-function studies that are already explained in the Introduction section. The authors should discuss the overexpression studies in greater detail, instead of simply mentioning that Sall4 is a “permissive” factor. Their data indicate that Sall4 overexpression alone is insufficient to evoke significant effects in their setting. They should discuss possible reasons based on the known molecular mechanisms of Sall4. They can compare their results with the previous reports with or without phenotypes; the findings may suggest technical differences or dependency on cellular context. The established mouse strain can also be used to address other biological questions. Such discussion will be informative for readers.

Reviewer #2: This manuscript reports the finding that over expression of both Sall4a and Sall4b do not affect the embryonic development and digit regeneration in the mouse model. The finding shows that, contrary to expectation, Sall4 over expression does not impart regenerative capacity to the limb, despite of its requirement for limb regeneration in amphibian animal models.

I reckon the work merits publication.

Your title emphasizes neonatal digit regeneration, but I am still very curious about whether you have performed P2 amputation experiment in adult stage. I understand that one would expect the non-regeneration of adult P2 phalanges when the neonatal P2 fail to regenerate with Sall4 over expression.

Another question is about Sall4a and Sall4b. You over-expressed both in your mouse models. Is there any difference between Sall4a versus Sall4b, in your expectation, to stimulate digit regeneration? The western results seem to suggest that expression of Sall4a is higher than Sall4b in the digit tip (with the same anti-Sall4). It was reported that Sall4b, but not Sall4a, "binds preferentially to highly expressed loci in ES cells" (doi: 10.1128/MCB.00419-10). You may want to discuss this.

Some suggestions:

1. Line 83, "The regenerative ability is level specific and restricted to the terminal

84 phalangeal bone", I suggest delete "bone".

2.Line 310, legend, "neonatal stage", I suggest indicating the time of digit amputation (P0). "neonatal stage" does not necessarily equal to day 0. For example, “The neonatal period is the first 4 weeks of a child's life".

6. PLOS authors have the option to publish the peer review history of their article (what does this mean?). If published, this will include your full peer review and any attached files.

Reviewer #1: No

Reviewer #2: No

---

## [Author Response · Author response to Decision Letter 0]

30 Mar 2022

Responses to the reviewers

Thank you for the critical and helpful comments regarding our manuscript (PONE-D-22-00272) entitled “Normal embryonic development and neonatal digit regeneration in mice overexpressing a stem cell factor, Sall4.”

We have examined the comments carefully and revised our manuscript. We believe that the we have adequately addressed the Reviewers’ concerns and improved the revised manuscript. Below, we provide point-by-point responses.

RESPONSE TO REVIEWER 1

Reviewer #1: Chen et al. inserted Sall4 into the Rosa26 locus and overexpressed Sall4 in mice via mesodermal Cre-dependent recombination. Although Sall4a and Sall4b were overexpressed in the caudal part of the embryo and the entire forelimb bud, minimal abnormalities were detected in early embryogenesis and digit regeneration. Although this manuscript describes negative results, the included information and the generated mouse strain are useful for the research community.

We thank the reviewer for considering “the included information and the generated mouse strain are useful for the research community”. Below, we provide response to each comment.

Major comments

Comment 1. The authors should describe the number of homologous recombinant ES clones and the number of independent mouse strains that were generated and analyzed. If this study involved only one ES clone and one mouse strain, those should be mentioned as limitations of the study; such information is necessary for objective interpretation by readers.

Response: Thank you for pointing out important information that was missing in the original manuscript. We picked up 64 ESC clones after G418 selection, and six of them were properly targeted by genomics PCR and Southern blotting. Among them, we used one clone for injection into the blastocyst after confirming normal karyotype. We have included this information as below (modified/new information is underlined)

Line 185-196: Among the 64 clones isolated after G418 selection, six properly targeted embryonic stem cell colonies were confirmed by genomic Southern blotting and genomic PCR (Fig 1C, D). After confirming normal karyotype, chimeric mice were generated by injecting one ES cell clone into blastocysts, which were bred with C57BL/6 mice. Germline transmission was confirmed by genomic PCR of offspring. This study was performed using this one strain.

Comment 2. Figure 2: A description should be provided concerning the number of biologically independent experiments performed with reproducible results. This point applies to all figures in the manuscript.

Response: Thank you for pointing this out. The number of biologically independent experiments/samples, which were not described are as below.

Fig. 2A-C (n=3), 2D-F (n=3), 2G, H (n=3)

Fig. 4A (n=4), 4B (n=3), 4C (n=3), 4D (n=3), 4E (n=3)

We have included this information in the main text.

Comment 3. Figure 2I: Because intensities differ among the loading controls (GAPDH bands), it is difficult to judge the quantitative increase in Sall4 protein expression. The ratio of Sall4 band intensity to GAPDH band intensity should be quantified, and the percentage increase should be provided. In addition, the results should be described in greater detail. For example, Sall4a overexpression is greater than Sall4b overexpression, while Sall4b expression is increased from baseline in the control.

Response: We have measured the Western blot band intensity by FIJI. The result shows that the ratio of SALL4A/GAPDH signal intensity was increased 1.7-fold in TCre+/Tg; R26-Sall4+/tg embryos, compared to TCre+/+; R26-Sall4+/tg embryos. We have included this information in the text as below.

Regarding SALL4B, it does not show detectable signals in the control sample. So, following the reviewer’s comment, we have included the following description. 

Line 239 – 243: Specifically, the data show that SALL4A overexpression is greater than SALL4B overexpression, while SALL4B expression is increased from baseline in the control. The ratio of SALL4A/GAPDH signal intensity was increased 1.7-fold in TCre+/Tg; R26-Sall4+/tg embryos, compared with TCre+/+; R26-Sall4+/tg embryos. Taken together, these results show that Sall4 is overexpressed by TCre-dependent recombination.

Comment 4. Figure 3I, J: the definitions of the dots and double dots should be clarified; they also appear to be mislocated in the graphs. Because there are no significant differences between Cre-WT/Rosa-Tg and Cre-Tg/Rosa-Tg, the effect of Sall4 overexpression is negligible. Instead, the presence of Sall4 cassette itself may mildly affect limb development. Such interpretation should be provided in the Results section, not the Discussion section.

Response: Thank you for pointing this out. The dots were introduced to indicate statistical significance. However, we indicated the p values in the panel, so we removed the dots from Fig 3I and J..

Regarding the interpretation of bone lengths, as suggested by the reviewer, we have included the following description.

Line 281 – 283: Given that the bone length between TCre+/+; R26-Sall4+/Tg and TCre+/Tg; R26-Sall4+/Tg did not show significant differences, the effect of Sall4 overexpression is negligible and the presence of the Sall4 cassette may mildly affect limb development.

Comment 5. The Discussion section largely consists of repetitive descriptions of the results; it also contains many descriptions of loss-of-function studies that are already explained in the Introduction section. The authors should discuss the overexpression studies in greater detail, instead of simply mentioning that Sall4 is a “permissive” factor. Their data indicate that Sall4 overexpression alone is insufficient to evoke significant effects in their setting. They should discuss possible reasons based on the known molecular mechanisms of Sall4. They can compare their results with the previous reports with or without phenotypes; the findings may suggest technical differences or dependency on cellular context. The established mouse strain can also be used to address other biological questions. Such discussion will be informative for readers.

Response: Thank you for this suggestion. We have extended our discussion as below. 

 First, we have discussed that Sall4 may require highly regenerative cellular or tissue context to contribute to limb regeneration (line 418-421).

 Second, we have discussed the possibility that SALL4 requires molecular partner to evoke significant effects in the tissue we examined (line 423 -433). 

 Third, we have discussed other biological questions that the Sall4 overexpression mouse strain may be useful (line 439 - 445).

 We hope that the reviewer acknowledges our new discussion.

RESPONSE TO REVIEWER 2

Reviewer #2: This manuscript reports the finding that over expression of both Sall4a and Sall4b do not affect the embryonic development and digit regeneration in the mouse model. The finding shows that, contrary to expectation, Sall4 over expression does not impart regenerative capacity to the limb, despite of its requirement for limb regeneration in amphibian animal models.

I reckon the work merits publication.

We thank the reviewer for his/her words “I reckon the work merits publication”. Below, we provide response to each comment.

Comment 1: Your title emphasizes neonatal digit regeneration, but I am still very curious about whether you have performed P2 amputation experiment in adult stage. I understand that one would expect the non-regeneration of adult P2 phalanges when the neonatal P2 fail to regenerate with Sall4 over expression.

Response: Thank you for this comment. As we showed in the manuscript, we did not observe regeneration after P2 amputation at the P0 stage. We think it is highly unlikely that the digit regenerates after P2 amputation in the adult stage, and therefore, we did not perform P2 amputation at the adult stage.

Comment 2: Another question is about Sall4a and Sall4b. You over-expressed both in your mouse models. Is there any difference between Sall4a versus Sall4b, in your expectation, to stimulate digit regeneration? The western results seem to suggest that expression of Sall4a is higher than Sall4b in the digit tip (with the same anti-Sall4). It was reported that Sall4b, but not Sall4a, "binds preferentially to highly expressed loci in ES cells" (doi: 10.1128/MCB.00419-10). You may want to discuss this.

Response: As the reviewer indicated, a previous study by the Orkin lab provided evidence that SALL4A and SALL4B have overlapping but non-identical binding sites in the mouse ES cell. However, we do not know functional differences between SALL4A and SALL4B in the Brachyury lineage during embryonic development and neonatal digit regeneration. Therefore, we simultaneously overexpressed both Sall4a and Sall4b in our study.

We have clarified this point in the Result as below (line: 177-180).

A previous study in mouse ES cells provided evidence that SALL4A and SALL4B have overlapping but non-identical binding sites (37). Because functional differences between SALL4A and SALL4B during mouse development are unknown, we expressed both Sall4a and Sall4b.

Some suggestions:

Comment 3 (Suggestion 1). Line 83, "The regenerative ability is level specific and restricted to the terminal phalangeal bone", I suggest delete "bone".

Response: Thank you for the suggestion. We have deleted “bone” from the sentence (new line 86).

Comment 4 (Suggestion 2) .Line 310, legend, "neonatal stage", I suggest indicating the time of digit amputation (P0). "neonatal stage" does not necessarily equal to day 0. For example, “The neonatal period is the first 4 weeks of a child's life".

Response: Thank you for the suggestion. We have indicated the amputation was performed at the P0 stage in line 344 in the revised manuscript.

---

## [Editor Report · Decision Letter 1]

6 Apr 2022

Normal embryonic development and neonatal digit regeneration in mice overexpressing a stem cell factor, Sall4

PONE-D-22-00272R1

Dear Dr. Kawakami,

We’re pleased to inform you that your manuscript has been judged scientifically suitable for publication and will be formally accepted for publication once it meets all outstanding technical requirements.

Kind regards,

Atsushi Asakura, Ph.D

Academic Editor

PLOS ONE
---

## [Editor Report · Acceptance letter]

8 Apr 2022

PONE-D-22-00272R1 

Normal embryonic development and neonatal digit regeneration in mice overexpressing a stem cell factor, *Sall4*

Dear Dr. Kawakami:

I'm pleased to inform you that your manuscript has been deemed suitable for publication in PLOS ONE. Congratulations! Your manuscript is now with our production department. 

Kind regards, 

on behalf of

Dr. Atsushi Asakura 

Academic Editor

PLOS ONE